# A Deep Learning Model for Cervical Optical Coherence Tomography Image Classification

**DOI:** 10.3390/diagnostics14182009

**Published:** 2024-09-11

**Authors:** Xiaohu Zuo, Jianfeng Liu, Ming Hu, Yong He, Li Hong

**Affiliations:** Department of Obstetrics and Gynecology, Renmin Hospital of Wuhan University, Wuhan 430060, China; 2021183029012@whu.edu.cn (X.Z.); liu_jianfeng@whu.edu.cn (J.L.); rena@whu.edu.cn (M.H.); yonghe@whu.edu.cn (Y.H.)

**Keywords:** cervical cancer, optical coherence tomography, computer-aided diagnosis, deep learning, multi-scale texture feature

## Abstract

**Objectives:** Optical coherence tomography (OCT) has recently been used in gynecology to detect cervical lesions in vivo and proven more effective than colposcopy in clinical trials. However, most gynecologists are unfamiliar with this new imaging technique, requiring intelligent computer-aided diagnosis approaches to help them interpret cervical OCT images efficiently. This study aims to (1) develop a clinically-usable deep learning (DL)-based classification model of 3D OCT volumes from cervical tissue and (2) validate the DL model’s effectiveness in detecting high-risk cervical lesions, including high-grade squamous intraepithelial lesions and cervical cancer. **Method:** The proposed DL model, designed based on the convolutional neural network architecture, combines a feature pyramid network (FPN) with texture encoding and deep supervision. We extracted, represent, and fused four-scale texture features to improve classification performance on high-risk local lesions. We also designed an auxiliary classification mechanism based on deep supervision to adjust the weight of each scale in FPN adaptively, enabling low-cost training of the whole model. **Results:** In the binary classification task detecting positive subjects with high-risk cervical lesions, our DL model achieved an 81.55% (95% CI, 72.70–88.51%) F1-score with 82.35% (95% CI, 69.13–91.60%) sensitivity and 81.48% (95% CI, 68.57–90.75%) specificity on the Renmin dataset, outperforming five experienced medical experts. It also achieved an 84.34% (95% CI, 74.71–91.39%) F1-score with 87.50% (95% CI, 73.20–95.81%) sensitivity and 90.59% (95% CI, 82.29–95.85%) specificity on the Huaxi dataset, comparable to the overall level of the best investigator. Moreover, our DL model provides visual diagnostic evidence of histomorphological and texture features learned in OCT images to assist gynecologists in making clinical decisions quickly. **Conclusions:** Our DL model holds great promise to be used in cervical lesion screening with OCT efficiently and effectively.

## 1. Introduction

Cervical cancer (CC) is one of the most common female malignant tumors that cause cancer-related death in women worldwide. Standard clinical screening approaches to CC include a Pap smear, thin-prep cytologic test (TCT), human papillomavirus (HPV) test, and colposcopy [1]. However, HPV tests have relatively high false-positive rates [2], while TCT and colposcopy have low sensitivity for cervical lesions [3]. Consequently, there is a pressing need for more accurate and efficient screening techniques to facilitate early detection and timely intervention.

Optical coherence tomography (OCT) [4] is a non-invasive, high-resolution, three-dimensional (3D) tomographic technique developed in the early 1990s. Since the mid-1990s, OCT has been used in ophthalmology, cardiology, and other medical fields. Recent clinical studies [1,5,6,7] have shown that OCT can detect cervical lesions more accurately than colposcopy by capturing histopathology-comparable morphological features of cervical tissue. However, the interpretation of cervical OCT images requires specialized expertise, which poses a significant challenge for widespread clinical adoption in gynecology.

In recent years, deep learning (DL) techniques have shown remarkable potential in medical image analysis, including diagnosing cervical diseases from cytology and colposcopy images [8,9,10,11,12,13]. Due to the inherent differences in imaging principles, OCT differs from digital pathology and colposcopy. Currently, there are no reported cases of directly applying existing classification models based on CNNs or vision transformers for cytopathology and colposcopy images to cervical OCT images using transfer learning. Recently, Ma et al. [14] proposed a two-stage approach based on a CNN-based feature extractor and a support vector machine classifier. They demonstrated the effectiveness of the proposed approach on an OCT volume dataset from 92 patients. However, this work did not consider texture analysis relevant to interpreting gray-scale medical images [15], such as computed tomography and magnetic resonance imaging, leaving room for improving classification performance.

This study aims to design an effective deep learning model to help gynecologists efficiently interpret OCT images from women undergoing cervical OCT tests, thereby addressing the challenge of accurately diagnosing cervical lesions. Our core solution is to extract and represent multi-scale texture features with a feature pyramid network (FPN) [16]. The technical contribution of our work is multi-scale texture encoding and integrated direct supervision for the DL model’s hidden layers to train the whole model at a low computational cost while providing interpretable representations of learned histomorphological and texture patterns. We will evaluate the performance of our DL model, trained with labeled OCT images from 699 patients in a multi-center clinical study, on two independent external validation datasets from two renowned hospitals in China. We will also compare it with experienced medical experts familiar with cervical OCT imaging modality to test diagnostic efficacy. Furthermore, our work provides understandable diagnostic interpretability based on histomorphological and texture features. This enables gynecologists to analyze cervical OCT volumes beyond simple image classification.

## 2. Materials and Methods

### 2.1. Data Collection

To train and validate the proposed DL model, we used an OCT image dataset from a multi-center clinical study conducted from August 2017 to December 2019 [1]. It contains more than 2800 3D cervical OCT volumes from 733 gynecological outpatients inspected with in-vivo OCT and received colposcopy-directed cervical biopsy. For details on the inspection process and OCT imaging, please refer to [1].

We collected two OCT image datasets of similar size from two renowned hospitals in China, Renmin Hospital of Wuhan University and West China Hospital of Sichuan University, to validate the performance of the proposed DL model in clinical trials. This study was approved separately by the Ethics Review Committee of the two hospitals. To be eligible to participate in this study, any patient with suspicious cervical diseases must meet the following requirements: (1) the subject is at least 18 years of age and older and sexually active; (2) the subject has not become pregnant; (3) the subject does not have a history of hysterectomy; and (4) the subject (or her legal guardians) must sign an informed consent form acknowledging the purpose of this study.

The Renmin dataset was collected at the Renmin Hospital of Wuhan University between February 2021 and May 2022. This dataset comprises TCT and HPV test results, cervical OCT volumes, and cervical biopsies from 113 gynecological patients. The second image dataset, the Huaxi dataset, includes 132 gynecological outpatients’ TCT and HPV test results, cervical OCT volumes, and cervical biopsies. It was collected between March 2021 and May 2021 at the West China Hospital of Sichuan University. Table 1 presents the demographic information of all of the subjects involved in this study.

### 2.2. OCT Image Processing

Each 3D cervical OCT volume is a tag image file format (TIFF) file that contains ten frames of two-dimensional OCT images obtained in vivo from an o’clock position on the cervix. The label of an OCT volume was determined by the formal diagnosis made for the corresponding biopsy specimen. However, high-resolution OCT volumes are too large to train CNN-based DL models efficiently. Two pathologists who have been using OCT for over one year extracted OCT images of 600 × 600 pixels from the original 3D volumes in the multi-center dataset and then annotated them. This process is similar to extracting digital pathology images from a whole-slide image.

These cervical OCT images, which matched well with the corresponding hematoxylin and eosin-stained pathology images, fell into five clinical categories: inflammation, cyst, ectropion, high-grade squamous intraepithelial lesion (HSIL), and CC. As with previous studies for binary classification [1,5,14], we classified mild inflammation (MI), ectropion (EP), and cyst (CY) as low risk (or negative) and HSIL and CC as high risk (or positive). Additionally, we designed an algorithm to automatically extract cervical OCT images using sliding windows from quantified OCT volumes without quality defects and sampling errors in the Remin and Huaxi datasets. Please refer to Figure A1 for details on the extraction process of cervical OCT images. Table 2 presents the statistics of the three OCT image datasets.

### 2.3. Model Design

In constructing a clinically usable DL model based on the CNN architecture, we consider two crucial factors neglected by previous works: multi-scale texture representation and efficient model training without complex attention mechanisms. We built an FPN with texture encoding to model multi-scale texture features from cervical OCT images. Our DL model’s feature encoding layer accepts arbitrary input sizes from several scales and learns visual vocabularies directly from each scale’s loss function. Inspired by the core idea of deeply supervised nets [17], we designed an auxiliary classification mechanism for different scales instead of self-attention to minimize classification error simultaneously. Backpropagation can be performed in an integrated, low-cost way to achieve the classification goal of high precision while reducing the prediction error at each scale.

#### 2.3.1. Model Architecture

As shown in Figure 1, the proposed DL model consists of three essential components: a backbone network, a feature encoding layer, and a classification layer. The in-network feature maps of four scales with the hierarchical structure are extracted from the backbone network. In the feature encoding layer, multi-scale texture features are obtained using the feature encoding module at each scale. In the classification layer, an auxiliary classifier generates a prediction result at each scale and fuses them into the final result. We will introduce the three components in the subsequent subsections.

#### 2.3.2. Backbone Network

The backbone network extracts feature maps from the input OCT image (a patch of an OCT volume) at four scales. Because it is a CNN model in this study, the network inputs were resized to 224 × 224 pixels to meet the size requirement of CNNs. The deep residual network (ResNet) [18] learns residual functions concerning the layer inputs and uses four stages containing bottleneck blocks to extract feature maps from the input images. A bottleneck block has three layers of 1 × 1, 3 × 3, and 1 × 1 convolutions and shortcut connections. We stack such blocks on top of each other to form a ResNet network. Stages near the top of ResNet are more likely to output large-size feature maps due to their larger receptive fields. As shown in Figure 1, the four ResNet stages generate feature maps of four different sizes, which are 56 × 56, 28 × 28, 14 × 14, and 7 × 7, respectively, to form an FPN.

#### 2.3.3. Texture Encoding

The feature encoding layer encodes the original feature maps extracted by the backbone network and generates a low-dimensional texture feature vector at each scale. This layer contains four branches corresponding to four scales. The four branches embed the original feature maps at different scales into texture feature spaces of the same dimension. As shown in Figure 2, each branch (namely the feature encoding module) comprises a convolutional block and an encoding block. The convolutional block contains a 1 × 1 convolution layer, batch normalization layer, and rectified-linear unit layer. It outputs a new feature map of 128 channels by reducing the dimension of the original feature map. The encoding block is an end-to-end module for texture analysis [19], which introduces the dictionary learning and feature pooling approach into the CNN pipeline. This block can learn the inherent visual vocabularies named texton [20] and convert the input feature map into a low-dimensional texture feature vector in the texton space.

Suppose ${X=\{x}_{i}|i=1,\cdots ,N\}$ is a set of image descriptors, $C=\{c_{k}|k=1,\cdots ,K\}$ is a dictionary with K learned textons (i.e., visual vocabularies), and the corresponding residual is $r_{ik}=x_{i}-c_{k}$. The aggregation operation for vector element $e_{k}$ can be represented as
(1)$$e_{k}=\sum_{i=1}^{N}a_{ik}r_{ik},$$
where $a_{ik}$ is the assigned weight calculated by
(2)$$a_{ik}=\frac{exp(-s_{k}{\left\|r_{ik}\right\|}^{2})}{\sum_{j=1}^{K}exp(-s_{j}{\left\|r_{ij}\right\|}^{2})},$$
where $s_{k}$ is the smoothing factor. Finally, the multi-scale feature encoding layer generates four texture feature vectors, each encoded by 32 textons of 128 dimensions.

#### 2.3.4. Deeply Supervised Classification

Texture features from four branches are flattened and fed to an auxiliary classifier to obtain predictions at different scales. The auxiliary classifier is composed of L2 normalization and a fully connected layer. These predictions are summed by element and fed to a softmax layer (namely the proposed DL model’s principal classifier) to generate a predicted label with the highest probability, the final classification result. This DL model’s loss function is mainly designed to ensure the supervision of the classification process at each scale and the final fusion process. Each auxiliary loss supervises the feature extraction and classification process at each scale, while the main loss evaluates the branching result fusion process to ensure the final classification performance.

More specifically, our DL model’s loss function consists of two parts (see Equation (3)): $L_{a}$ is an auxiliary loss corresponding to the auxiliary classifier at a scale and $L_{m}$ is the main loss corresponding to the principal classifier. In the fusion process, the DL model achieves the adaptive adjustment of weights by converging the loss functions. Moreover, we added the label smoothing regularization to improve the model’s robustness while calculating the loss function.
(3)$$Loss={\sum L_{a}\left({\hat{y}}_{a},y\right)+L}_{m}({\hat{y}}_{m},y),$$
where ${\hat{y}}_{a}$ and ${\hat{y}}_{m}$ represent the prediction of the auxiliary classifier and the principal classifier, respectively, and y is the actual label of the input image. The cross-entropy loss function (see Equation (4)) evaluates the difference between predicted and ground truth labels.
(4)$$L\left(\hat{y},y\right)=-\sum_{i=1}^{C}y_{i}log({\hat{y}}_{i}),$$
where C represents the total number of categories, $y_{i}$ denotes the value of one-hot coded label on dimension i, and ${\hat{y}}_{i}$ is the corresponding prediction on the dimension.

#### 2.3.5. OCT Volume Label Prediction

Because the DL model only outputs the label of a given cervical OCT image, we need to predict an entire volume’s category by aggregating the prediction results of all patches it contains. Despite the existence of the majority voting rule and machine learning-based approaches, we used a cross-shaped threshold voting rule [21] to interpret patch-level predictions. That is to say, a cervical lesion in a 3D OCT volume can be detected in consecutive high-risk patches defined by a threshold from the cross-sectional (in-frame) and axial (cross-frame) directions. Such a pattern looks like a cross, which is straightforward and explainable for gynecologists. Figure A2 illustrates an OCT volume of HSIL identified using the cross-shaped threshold voting rule. As with colposcopy-directed biopsy, a subject is positive (or high risk) if one of her cervical OCT volumes is diagnosed as positive.

### 2.4. Experiment Setups

#### 2.4.1. Comparison Experiment

As shown in Figure 3, we designed two comparison experiments: machine–machine and human–machine comparisons. The machine–machine comparison aims to validate the effectiveness of our DL model internally, which uses a commonly used CNN model, ResNet-101 [18], as the backbone network. Moreover, we utilized the ten-fold cross-validation method to split the multi-center dataset into ten folds according to the patient identification number. Cervical OCT images from the same patient cannot be involved simultaneously in the training and test sets to avoid over-fitting. As anonymous investigators, five medical experts familiar with OCT from different hospitals, including two pathologists, one radiographer, and two gynecologists, were invited to participate in the human–machine comparison to diagnose cervical OCT volumes from the Remin and Huaxi datasets.

#### 2.4.2. Data Preprocessing for Model Training

In the model training phase, each input OCT image was resized to 224 × 224 pixels required by the CNN architecture and then normalized using the z-score normalization method. We also leveraged data augmentation techniques to enhance the richness of the training data. Specifically, the training data were enriched by randomly flipping the input image horizontally or vertically, adding noise, and adjusting the input image’s brightness. A weighted random oversampling method was also adopted to solve the class imbalance problem of cervical OCT images. It generated new samples in the under-represented class (i.e., CC).

#### 2.4.3. Parameter Settings

The label smoothing parameter E was set to 0.1, which means the predicted label has a 90% probability of following the original distribution and a 10% probability of conforming to the uniform distribution. During the training process, the batch size was set to 16, the total number of epochs was set to 30, and the backbone network was configured with its default setting. CNN-based models were trained using the stochastic gradient descent optimizer with momentum [22], with an initial learning rate of 0.005 and a momentum value of 0.9. The learning rate was decayed within the training process of 30 epochs with cosine annealing for each batch. In all feature encoding modules, the number of textons K was set to 32, and the dimension of textons D was set to 128. The threshold value of the cross-shaped threshold voting rule for high risk (or positive) was set to 0.8 when predicting the label of an OCT volume. For more details on the model implementation and source code, please refer to the code repository at https://github.com/WanrongDou/MSE (accessed on 20 August 2024).

#### 2.4.4. Evaluation Metrics

In addition to three commonly used evaluation measures, namely accuracy, sensitivity, and specificity, positive predictive value (PPV) and negative predictive value (NPV) were employed to assess the prediction performance of different CNN-based classification models and investigators in the two comparison experiments. Specifically, in the machine-machine comparison of images, the overall performance of a model was evaluated in terms of the area under the curve (AUC). In the human–machine comparison of patients, we evaluated the overall performance of the proposed DL model and investigators for 3D OCT volumes according to the F1 score. Please refer to Section A.3 for the definitions of these evaluation metrics.

### 2.5. Statistical Analysis

We used the Mann–Whitney U Test, a nonparametric hypothesis test that compares two independent groups, to perform a statistical analysis of our experimental results. This hypothesis test determines whether the difference between the medians of the two groups is statistically significant, especially for the human–machine comparison in our study.

## 3. Results

### 3.1. Machine–Machine Comparison with Baseline Models

Table 3 shows the image-level classification results of different CNN-based models in five-class and binary classification tasks. Numbers in bold indicate the best results in terms of evaluation metrics. B-TE denotes a model built on the backbone network and single-scale texture encoding, and B-FPN denotes a base model composed of the backbone network and an FPN. B-F-T and B-F-C represent two baseline models that combine the B-FPN model with multi-scale texture encoding and the deeply supervised classifier; more specifically, they are two partially ablated models of our DL model.

Our DL model performed best in the five-class classification task and achieved an accuracy of 88.67 ± 2.94%, followed by B-F-C with 87.54 ± 3.82% accuracy. All of the CNN-based models performed well in the binary classification task because their AUC values exceeded 0.97. Our DL model achieved the highest AUC value of 0.9880 ± 0.005 with a 92.70 ± 3.85% sensitivity and 96.28 ± 2.23% specificity. In particular, it achieved a PPV of 94.21 ± 4.65%, 1.24% higher than that of B-F-C. This result indicates the more significant ability of our model with two necessary components to identify true positives than other baseline models.

### 3.2. Human–Machine Comparison with Medical Experts

Table 4 presents the patient-level prediction results of our DL model and five medical experts in the binary classification task that detects high-risk cervical lesions. Here, the F1-score metric evaluates the overall performance of the DL model and investigators for cervical OCT volumes. Confidence intervals (CIs) for the six evaluation metrics are exact Clopper–Pearson confidence intervals at the 95% confidence level [23]. Avg. denotes the average level of the five investigators.

Although the investigators’ performance differs in the two datasets, our DL model has good diagnostic consistency with the F1-score value greater than 80% in the binary classification task. As shown in Table 4, the second and third investigators performed best on the Renmin and Huaxi datasets regarding the F1-score metric. Our DL model achieved 81.91% accuracy and an 81.55% F1-score on the Renmin dataset, 6.67% and 12.50% higher than that of the best performer (the second investigator), respectively. Also, our DL model’s overall performance was comparable to that of the best performer (the third investigator) on the Huaxi dataset. Its values of accuracy and F1-score were only 2.40% and 2.15% lower than that of the best performer, respectively. In particular, our DL model achieved the highest sensitivity value on the two datasets, significantly defeating the five investigators (i.e., there was a statistically significant difference between the model and each investigator, *p*-value < 0.01). This result indicates that the model can effectively identify high-risk cervical diseases using texture features extracted from OCT images.

Figure 4 displays the confusion matrices for the classification performance of our DL model and the five investigators in the binary classification task. Such a confusion matrix visually describes how many patients are correctly and incorrectly classified for each general category (positive or negative). Our DL model performed best on the Renmin dataset, making only nineteen misclassifications, including ten false positives and nine false negatives. Instead, the five investigators had higher rates of missed diagnosis for high-risk cervical lesions. On the Huaxi dataset, the five investigators and our DL model made fewer misclassifications because of the higher image quality of the collected cervical OCT volumes. Although our DL model made three more misclassifications compared with the best performer (the third investigator), it achieved a lower rate of false negatives than the five investigators. The promising results show that the model holds great potential to assist gynecologists in identifying high-risk cervical lesions in the clinical environment.

### 3.3. Interpretability Study

The diagnostic interpretability of DL models has always been a concern in medical image analysis. Doctors must understand how a DL model makes judgments for an input medical image. In this study, we leverage an improved version of the class activation map (CAM), Grad-CAM [24], to highlight those regions containing histomorphological and texture features extracted by our DL model and provide diagnostic evidence for gynecologists to detect cervical diseases better. Figure 5 presents five typical examples of cervical tissue, each with an OCT image and four heatmaps generated by Grad-CAM at different scales. Different image features are extracted from the original OCT image at four scales. The general features of the whole OCT image are extracted at the first scale (Scale I). Along with the decrease in scale, the discriminative feature information is aggregated from the above scales at the last scale (Scale IV).

As shown in Figure 5, our DL model concentrated on the region proximal to the basement membrane (BM) in the CAM image (the first row and fifth column), indicating a normal gynecological case with cervical inflammation. It made such a decision because a well-defined hierarchical structure and distinct BM existed, a typical histomorphological feature identified by pathologists [5]. Regarding the cyst image (the second row and fifth column), the model focused on the epithelial tissue and cystic areas, providing a negative diagnostic outcome based on the above histomorphological feature. Extraversion of the columnar epithelium (also called cervical ectropion) is a normal state of the cervix, but it often lacks a hierarchical structure in OCT images and can be misdiagnosed as CC. The model emphasized the papillary structure on the left side of the CAM image (the third row and fifth column) to assist gynecologists in reaching a negative diagnosis. In previous studies [1,5,14], hyper-scattering papillary structures have been identified as a typical histological feature for EP. For positive (HSIL and CC) cases, the hierarchical structure was absent from the OCT images, and BM was indistinguishable too. The model directed attention to the lesion or tumor area, characterized by alternating light and dark regions due to light attenuation [1,5,14], enabling reasonable diagnostic assessments.

## 4. Discussion

The proposed DL model for classifying 3D OCT volumes of cervical tissue demonstrates promising results in detecting HSIL and CC from OCT images. By leveraging an FPN with texture encoding and deep supervision, our model effectively captures multi-scale texture features and learns discriminative representations for accurate cervical tissue classification.

One notable limitation of our study is the lack of a comprehensive analysis of potential sources of bias and factors that may have influenced the model’s performance on external validation datasets. While we observed comparable or superior performance to experienced medical professionals, we did not critically examine the potential impact of differences in data quality, patient demographics, or imaging protocols across different medical centers. These factors could potentially introduce biases and affect the generalizability of our model’s predictions. Therefore, future studies should investigate the robustness of the proposed DL model to variations in data acquisition and preprocessing methods and explore techniques for mitigating potential biases. Additionally, prospective multi-center clinical trials with larger and more diverse patient cohorts would be valuable to further validate the model’s efficacy and generalizability across different healthcare settings and populations.

Despite this limitation, our work provides a promising foundation for developing computer-aided diagnosis systems for CC screening using OCT imaging. Compared with a recent study [14], our model incorporated multi-scale texture features and deep supervision, and improved classification performance. By offering interpretable diagnostic evidence based on learned histomorphological and texture features, our model can assist gynecologists in making informed clinical decisions and potentially improve the efficiency and accuracy of traditional cervical lesion screening, such as ThinPrep Cytology combined with HPV.

We selected low-risk, pathology-negative patients with positive HPV results, 44 from the Renmin dataset and 74 from the Huaxi dataset. Our DL model and medical experts achieved considerable accuracy in only interpreting cervical OCT volumes to identify low-risk patients (from the model to the five investigators): 84.09%, 86.36%, 95.45%, 90.91%, 100%, and 63.64% (investigators’ average: 87.27%) on the Renmin dataset and 89.19%, 90.54%, 95.95%, 100%, 90.54%, and 83.78% (investigators’ average: 92.16%) on the Huaxi dataset. This result suggests that OCT can reduce the number of excessive colposcopy and cervical biopsies for HPV-positive patients, which is consistent with the conclusion of recent studies [25,26]. Moreover, we collected high-risk, pathology-positive patients with negative TCT results, 31 from the Renmin dataset and 29 from the Huaxi dataset. Our DL model achieved 90.32% and 82.76% accuracy in identifying high-risk patients with only cervical OCT volumes on the two datasets, higher than those of the five investigators (35.48%, 48.39%, 38.71%, 25.81%, and 61.29% on the Remin dataset (*p*-value < 0.001) and 79.31%, 79.31%, 75.86%, 55.17%, and 65.52% on the Huaxi dataset (*p*-value < 0.01)). This result indicates that our DL model has the advantage of detecting high-risk cervical lesions in real-time over TCT.

Furthermore, integrating our DL model into clinical workflows could facilitate the broader adoption of OCT as a screening modality, bridging the gap between advanced imaging techniques and routine clinical practice, and could potentially improve patient outcomes. However, it is crucial to note that our model should be utilized as a complementary tool to support human experts rather than as a standalone diagnostic system until further validation and regulatory approvals are obtained.

In conclusion, our study demonstrates the potential of DL techniques for enhancing CC screening by analyzing 3D OCT volumes. By addressing the identified limitations and conducting further research, we can refine and strengthen our model, ultimately contributing to improved patient outcomes and reduced CC-related morbidity and mortality.

## 5. Conclusions

CC is one of the most common female malignant tumors worldwide. However, standard clinical screening approaches, including HPV and TCT tests, have many limitations. Our DL model, elaborately designed by combining FPN with texture encoding and deep supervision, holds great promise in being used in cervical lesion screening with OCT efficiently and effectively. The experimental results indicate that the model can outperform other baseline models in binary and five-class classification tasks and achieve classification results close to (or even better than) the average level of five medical experts with better interpretability based on histomorphological and texture features, which holds great potential to assist gynecologists in identifying high-risk cervical lesions in the clinical environment.

## Figures and Tables

**Figure 1 diagnostics-14-02009-f001:**
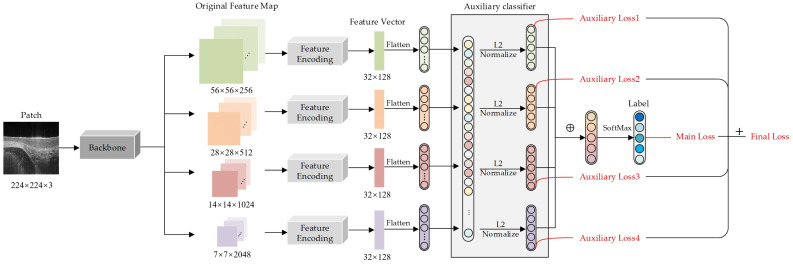
Model architecture. In addition to a backbone network, we elaborately designed an FPN with texture encoding and an auxiliary classifier in our DL model. The backbone network extracts feature maps of different scales from the input OCT images. The texture encoding layer learns visual vocabularies directly from each scale’s loss function and generates a feature vector at each scale. The main loss of our DL model comprises each scale’s auxiliary loss and the principal classifier’s loss.

**Figure 2 diagnostics-14-02009-f002:**
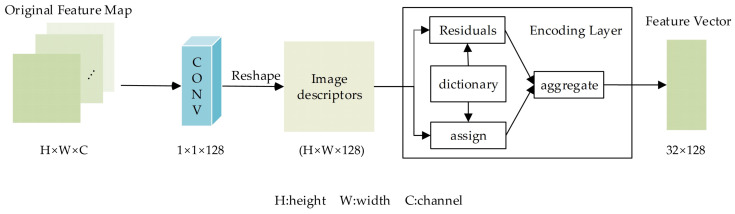
Feature encoding module. It contains a convolutional block and an encoding block. The convolutional block converts the original feature map of H×W×C into a set of image descriptors of (H×W)×128. Then, the encoding layer, which leverages dictionary learning and feature pooling, maps the image descriptors into a low-dimensional feature vector at each scale.

**Figure 3 diagnostics-14-02009-f003:**
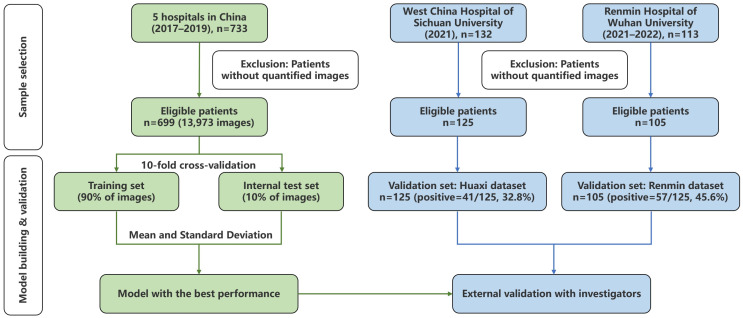
Workflow diagram of comparison experiments. We chose the best-performing DL model through ten-fold cross-validation on the multi-center dataset. Then, we tested this model’s diagnostic efficacy on two external datasets by comparing it with five medical experts.

**Figure 4 diagnostics-14-02009-f004:**
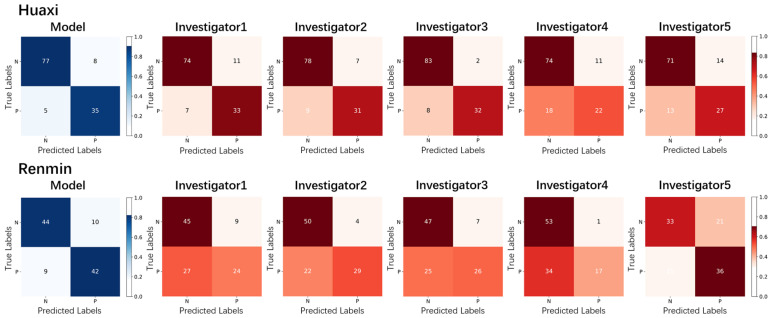
Confusion matrices of our DL model and five investigators in the binary classification task detecting high-risk cervical lesions on two external validation datasets.

**Figure 5 diagnostics-14-02009-f005:**
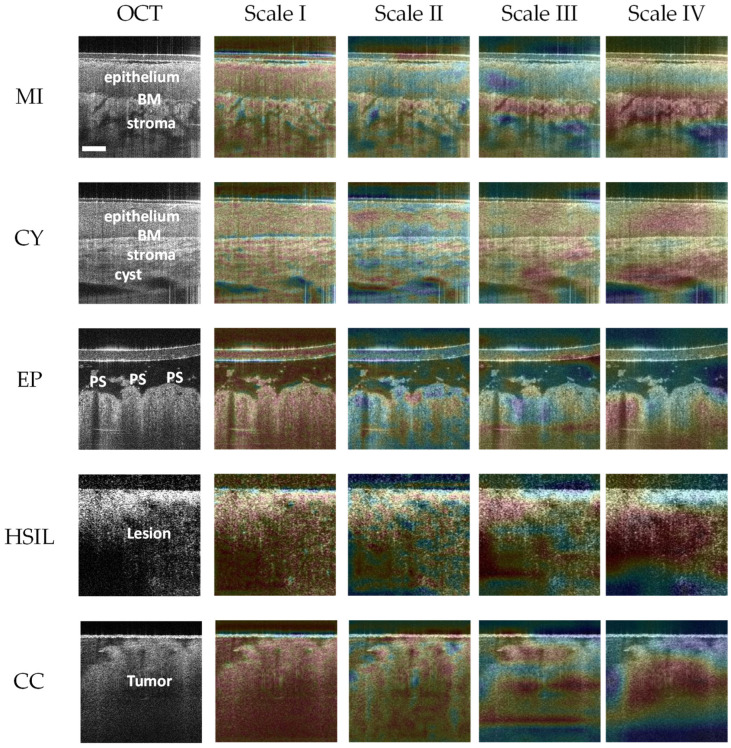
Pixel-level histomorphological and texture feature visualization at four scales. Scale I and IV correspond to 56 × 56 and 7 × 7, respectively. The discriminative features of cervical OCT images are aggregated at the last scale (i.e., Scale IV). We overlay the Scale-IV heatmaps representing the DL model’s attention on the original images to assist gynecologists in interpreting cervical OCT images. BM: basement membrane; PS: papillary structures with hyper-scattering boundaries; scale bar: 200 μm.

**Table 1 diagnostics-14-02009-t001:** Demographic information of subjects.

Dataset	Hospital	Number	Age	HPV Results	TCT Results
Multi-center	The Third Affiliated Hospital of Zhengzhou University	350	38.67 ± 9.86	Positive: 281 Negative: 31 Untested: 38	Positive: 215 Negative: 63 Untested: 72
	Liaoning Cancer Hospital and Institute	227	44.08 ± 8.38	Positive: 138 Negative: 84 Untested: 5	Positive: 69 Negative: 134 Untested: 24
	Puyang Oilfield General Hospital	59	43.04 ± 8.06	Positive: 39 Negative: 3 Untested: 17	Positive: 39 Negative: 12 Untested: 8
	Luohe Central Hospital	57	40.37 ± 10.05	Positive: 49 Negative: 8 Untested: 0	Positive: 36 Negative: 21 Untested: 0
	Zhengzhou Jinshui District General Hospital	40	39.03 ± 12.47	Positive: 36 Negative: 2 Untested: 2	Positive: 9 Negative: 27 Untested: 4
	Overall	733	40.85 ± 9.79	Positive: 543 Negative: 128 Untested: 62	Positive: 368 Negative: 257 Untested: 108
Renmin	Renmin Hospital of Wuhan University	113	49.68 ± 11.77	Positive: 98 Negative: 12 Untested: 3	Positive: 20 Negative: 74 Untested: 19
Huaxi	West China Hospital of Sichuan University	132	39.02 ± 9.09	Positive: 116 Negative: 7 Untested: 9	Positive: 16 Negative: 112 Untested: 4

**Table 2 diagnostics-14-02009-t002:** A brief introduction to the experimental datasets.

Dataset	Size	MI	CY	EP	HSIL	CC	Total
Multi-center	# Patients	239	126	99	161	74	699
	# Volumes	363	195	153	166	379	1256
	# Images	3172	2464	2067	5539	731	13,973
Remin	# Patients	20	18	10	36	21	105
	# Volumes	241	216	121	432	254	1264
Huaxi	# Patients	40	27	17	39	2	125
	# Volumes	301	216	148	87	8	760

**Table 3 diagnostics-14-02009-t003:** Performance comparison of the CNN-based classification models (mean ± std).

	Backbone	B-TE	B-FPN	B-F-T	B-F-C	Ours
Acc5	84.56 ± 4.02%	86.10 ± 3.41%	85.77 ± 3.92%	85.94 ± 3.13%	87.54 ± 3.82%	**88.67 ± 2.94%**
Acc2	92.39 ± 2.04%	93.72 ± 1.59%	92.97 ± 2.80%	93.38 ± 1.38%	94.52 ± 1.73%	**94.89 ± 1.52%**
Sens	89.56 ± 5.46%	91.96 ± 2.84%	91.11 ± 4.35%	91.35 ± 4.36%	**93.06 ± 3.65%**	92.70 ± 3.85%
Spec	93.98 ± 2.43%	94.88 ± 2.93%	94.05 ± 3.49%	94.49 ± 2.82%	95.42 ± 2.62%	**96.28 ± 2.23%**
PPV	90.84 ± 5.22%	92.55 ± 5.17%	91.03 ± 6.47%	92.14 ± 4.63%	92.97 ± 4.76%	**94.21 ± 4.65%**
NPV	92.96 ± 2.26%	94.13 ± 2.62%	93.93 ± 2.79%	94.02 ± 2.76%	**95.24 ± 2.33%**	95.06 ± 2.30%
AUC	97.09 ± 1.50%	98.04 ± 1.09%	97.95 ± 1.32%	98.12 ± 0.71%	98.72 ± 0.56%	**98.80 ± 0.50%**

Acc2: accuracy for binary classification tasks. Acc5: accuracy for five-class classification tasks. Sens: sensitivity. Spec: specificity. The bolded numbers in the table represent the optimal values.

**Table 4 diagnostics-14-02009-t004:** Performance comparison between two human experts and our model (95%CI).

Dataset		Acc2 (%)	Sens (%)	Spec (%)	PPV (%)	NPV (%)	F1-Score (%)
Renmin	Investigator 1	65.71	47.06	83.33	72.73	62.50	57.14
		(55.81–74.70)	(32.93–61.54)	(70.71–92.08)	(54.48–86.70)	(50.30–73.64)	(45.88–67.89)
	Investigator 2	75.24	56.86	92.59	87.88	69.44	69.05
		(65.86–83.14)	(42.25–70.65)	(82.11–97.94)	(71.80–96.60)	(57.47–79.76)	(58.02–78.69)
	Investigator 3	69.52	50.98	87.04	78.79	65.28	61.91
		(59.78–78.13)	(36.60–65.25)	(75.10–94.63)	(61.09–91.02)	(53.14–76.12)	(50.66–72.29)
	Investigator 4	66.67	33.33	**98.15**	**94.44**	60.92	49.28
		(56.80–75.57)	(20.76–47.92)	(90.11–99.95)	(72.71–99.86)	(49.87–71.21)	(37.02–61.59)
	Investigator 5	65.74	70.59	61.11	63.16	68.75	66.67
		(55.81–74.70)	(56.17–82.51)	(46.88–74.08)	(49.34–75.55)	(53.75–81.34)	(56.95–75.45)
	Avg. (95% CI)	68.57	51.77	84.44	79.40	65.38	60.81
		(58.78–77.28)	(37.34–65.98)	(72.01–92.87)	(58.40–88.69)	(52.64–75.97)	(50.41–71.85)
	Ours	**81.91**	**82.35**	81.48	80.77	**83.02**	**81.55**
		(73.19–88.74)	(69.13–91.60)	(68.57–90.75)	(67.47–90.37)	(70.20–91.93)	(72.70–88.51)
Huaxi	Investigator 1	85.60	82.50	87.06	75.00	91.36	78.57
		(78.20–91.24)	(67.22–92.66)	(78.02–93.36)	(59.66–86.81)	(83.00–96.45)	(68.26–86.78)
	Investigator 2	87.20	77.50	91.76	81.58	89.66	82.67
		(80.05–92.50)	(61.55–89.16)	(83.77–96.62)	(65.67–92.26)	(81.27–95.16)	(68.84–87.80)
	Investigator 3	**92.00**	80.00	**97.65**	**94.12**	91.21	**86.49**
		(85.78–96.10)	(64.35–90.95)	(91.76–99.71)	(80.32–99.28)	(83.41–96.13)	(76.55–93.32)
	Investigator 4	76.80	55.00	87.06	66.67	80.43	60.27
		(68.41–83.88)	(38.49–70.74)	(78.02–93.36)	(48.17–82.04)	(70.85–87.97)	(48.14–71.55)
	Investigator 5	78.40	67.50	83.53	65.85	84.52	66.67
		(70.15–85.26)	(50.87–81.43)	(73.91–90.69)	(49.41–79.92)	(74.99–91.49)	(55.32–76.76)
	Avg. (95% CI)	84.00	72.50	89.41	76.64	87.44	74.93
		(76.38–89.94)	(56.11–85.40)	(80.85–95.04)	(59.76–88.56)	(78.50–93.52)	(63.21–83.58)
	Ours	89.60	**87.50**	90.59	79.55	**93.90**	84.34
		(82.87–94.35)	(73.20–95.81)	(82.29–95.85)	(66.60–91.61)	(86.34–97.99)	(74.71–91.39)

Acc2: accuracy for binary classification tasks. Sens: sensitivity. Spec: specificity. The bolded numbers in the table represent the optimal values.

## Data Availability

The three cervical OCT image datasets used in this study are not publicly available due to privacy or ethical restrictions, but typical examples are available from the corresponding authors upon request.

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
