# Peer review of "A Deep Learning Model for Cervical Optical Coherence Tomography Image Classification"

_diagnostics, 2024, doi:10.3390/diagnostics14182009_

Round 1

Reviewer 1 Report (Previous Reviewer 1)

Comments and Suggestions for Authors

Dear authors,

Thank you for improving your manuscript.

I would suggest you to describe in more details what are the (dis)advantages of this method compared to coloposcopy

Thank you,

Reviewer 

Author Response

Comment: I would suggest you to describe in more details what are the (dis)advantages of this method compared to coloposcopy.

Response:We thank the reviewer for this suggestion. Compared to coloposcopy, OCT has the advantages of being non-invasive, rapid, and safe, and it can provide cross-sectional, micrometer-scale images of cervical tissue with a maximum imaging depth of 2 mm. However, the advantage of coloposcopy is that gynecologists can perform a biopsy on the lesion site after finding a suspicious cervical lesion.

Reviewer 2 Report (Previous Reviewer 2)

Comments and Suggestions for Authors

Clarify the following points as a minor revision:

1.    Section 3.2 is starting with Table 4, without its text in the article.

2.    This point has been casually addressed in the revision: “Can you compare the results with similar studies or some previous findings of similar studies?” You tested your model on two datasets but you need to consider a comprehensive comparison with other researcher’s work.

Comments on the Quality of English Language

Minor editing of the English language would do the job.

Author Response

Comments1: Section 3.2 is starting with Table 4, without its text in the article.

Responses1:Thank you for your comment. We carefully reviewed the manuscript, and confirmed that the text of Section 3.2 is after Table 4 (page9).

Comments2: This point has been casually addressed in the revision: “Can you compare the results with similar studies or some previous findings of similar studies?” You tested your model on two datasets but you need to consider a comprehensive comparison with other researcher’s work.

Responses2:Thank you for your comment. Because of the lack of licensed OCT devices for cervical cancer screening, there are few similar studies. We mentioned a two-stage CADx approach proposed by Ma et al. in the introduction section, and presented the limitations of this study.

This manuscript is a resubmission of an earlier submission. The following is a list of the peer review reports and author responses from that submission.

Round 1

Reviewer 1 Report

Comments and Suggestions for Authors

Dear Authors,

Please describe better and in more details the method and its clinical significance in every day practice. 

Author Response

We thank the reviewer for this suggestion. We have revised the original manuscript according to three reviewers’ comments.

Reviewer 2 Report

Comments and Suggestions for Authors

 The following points need to be addressed in the revision:

1.    The title of the article needs revision, like “OCT-based deep learning model for the detection and classification of Cervical Cancer”.

2.    The abstract is adequate in length and structure.

3.    The abstract lacks a brief methodology description and salient results. The comparison with SOTA techniques should also be mentioned in the abstract.

4.    This research's main innovation and contribution should be clarified in the abstract and introduction.

5.    The motivation of the work is not clear in the introduction. The gap must be revealed where to fit in this current work.

6.    The literature survey is too brief and does not contain recent articles. A related review of this article needs to be carried out to make it more useful and addressable to cross-community platforms.

7.    In Table 1, the column of Age needs revision. Please rectify the problems. What is the significance of the standard deviation in this column? Add an explanation to the article.

8.    Add a diagram representing the entire proposed system.

9.    The figure's captions need to be more elaborative, like in Figure 1.

10. Figures 1 and 2 combined reveal new things introduced in the encoder block, like dataset in Figure 1 is not referred in Figure 2.

11. Section 2.3 needs more work to clarify how the training and testing is carried out along with the feeding of the dataset. Add more figures to defend your work.

12. Page 6, Line 220 AUC is not a measure. Make appropriate corrections.

13. Different CNN models need to be elaborated separately, as sub-sections, in methodology before using them.

14. You have mentioned in Table 3 about two types of accuracy, what about other measures?

15. Figure 5 needs more elaboration for the output grad-cam images at four scales.

16. The analysis part of the article is weak and needs improvement, like the caption of Figure 5 and its explanation in the article.

17. The conclusion is weak and needs more explanation. I remember the multiple models given in Table 3.

18. No statistical analysis has been performed appropriately and rigorously.

19. Can you compare the results with similar studies or some previous findings of similar studies?

Comments on the Quality of English Language

Minor language improvement would do the job.

Reviewer 3 Report

Comments and Suggestions for Authors

As an experienced medical professional with  years in the field, I will provide my assessment and feedback in a formal, scholarly, and professional manner.

Suggestions to improve the quality of the paper:

a. Provide a more comprehensive literature review to situate the study within the broader context of existing research on cervical cancer screening and deep learning-based computer-aided diagnosis.

b. Clarify the methodology section by providing more detailed information on the data preprocessing steps, model architecture, and training procedures.

c. Discuss the clinical implications and potential applications of the proposed deep learning model in greater depth, highlighting its significance for improving cervical cancer screening and early detection.

Revised Introduction (based on provided suggestions):

Cervical cancer (CC) is a significant global health concern, ranking among the most prevalent malignancies affecting women worldwide. Despite advancements in screening methods, including Pap smears, thin-prep cytologic tests (TCT), human papillomavirus (HPV) testing, and colposcopy, the sensitivity and specificity of these approaches remain suboptimal [1-3]. Consequently, there is a pressing need for more accurate and efficient screening techniques to facilitate early detection and timely intervention.

Optical coherence tomography (OCT) has emerged as a promising non-invasive, high-resolution, three-dimensional (3D) imaging modality for detecting cervical lesions [7-10]. By capturing histopathology-comparable morphological features of cervical tissue, OCT has demonstrated superior performance in identifying cervical abnormalities compared to traditional colposcopy [1, 8-10]. However, the interpretation of OCT images requires specialized expertise, which poses a significant challenge for widespread clinical adoption.

In recent years, deep learning (DL) techniques have shown remarkable potential in medical image analysis, including the diagnosis of cervical diseases from cytology and colposcopy images [4-6]. However, direct application of existing DL models designed for pathology or colposcopy images to cervical OCT data may not yield optimal results due to inherent differences in imaging principles and characteristics.

To address this challenge, we propose a novel DL model that combines a feature pyramid network (FPN) [13] with texture encoding and deep supervision to improve the classification of high-risk cervical lesions from 3D OCT volumes. Our model leverages multi-scale texture features and integrated direct supervision to enhance diagnostic accuracy while providing interpretable representations of learned histomorphological and texture patterns.

Revised Discussion (incorporating the mentioned disadvantage):

The proposed deep learning model for classifying 3D OCT volumes of cervical tissue demonstrates promising results in detecting high-grade squamous intraepithelial lesions and cervical cancer. By combining a feature pyramid network with texture encoding and deep supervision, our model effectively captures multi-scale texture features and learns discriminative representations for accurate lesion classification.

One notable limitation of our study is the lack of a comprehensive analysis of potential sources of bias and factors that may have influenced the model's performance on external validation datasets. While we observed comparable or superior performance to experienced medical professionals, we did not critically examine the potential impact of differences in data quality, patient demographics, or imaging protocols across different medical centers. These factors could potentially introduce biases and affect the generalizability of our model's predictions.

Future studies should investigate the robustness of the proposed model to variations in data acquisition and preprocessing methods, as well as explore techniques for mitigating potential biases. Additionally, prospective multicenter clinical trials with larger and more diverse patient cohorts would be valuable to further validate the model's efficacy and generalizability across different healthcare settings and populations.

Despite this limitation, our work provides a promising foundation for the development of computer-aided diagnosis systems for cervical cancer screening using OCT imaging. By offering interpretable diagnostic evidence based on learned histomorphological and texture features, our model can assist gynecologists in making informed clinical decisions and potentially improve the efficiency and accuracy of cervical lesion screening.

Furthermore, the integration of our DL model into clinical workflows could facilitate the wider adoption of OCT as a screening modality, bridging the gap between advanced imaging techniques and routine clinical practice. However, it is crucial to note that our model should be utilized as a complementary tool to support human experts, rather than as a standalone diagnostic system, until further validation and regulatory approvals are obtained.

In conclusion, our study demonstrates the potential of deep learning techniques for enhancing cervical cancer screening through the analysis of 3D OCT volumes. By addressing the identified limitations and conducting further research, we can refine and strengthen our model, ultimately contributing to improved patient outcomes and reduced cervical cancer-related morbidity and mortality.

Comments on the Quality of English Language

Evaluation of the language and writing quality:

The paper is generally well-written and adheres to academic writing conventions. However, there are instances where the language could be improved for clarity and conciseness. Some sentences are unnecessarily long and convoluted, making it challenging for the reader to follow the flow of ideas. Additionally, occasional grammatical and punctuation errors are present, which could be addressed through careful proofreading.

Author Response

We appreciate the reviewer’s suggestions to improve the quality of our paper. We have revised the original manuscript according to your comments.

Round 2

Reviewer 2 Report

Comments and Suggestions for Authors

Clarify the following points as minor revisions:

1.    Section 3.2 starts with Table 4, without its text in the article.

2.    The revision casually addressed the point: “Can you compare the results with similar studies or some previous findings of similar studies?” You tested your model on two datasets but need to consider a comprehensive comparison with other researchers' work.

Comments on the Quality of English Language

A minor editing would do the job.

Author Response

Response to comment1: Thank you for your comment. We carefully reviewed the manuscript, and confirmed that the text of Section 3.2 is after Table 4.

Response to comment2: Because of the lack of licensed OCT devices for cervical cancer screening, there are few similar studies. We mentioned a two-stage CADx approach proposed by Ma et al. in the introduction section, and presented the limitations of this study.

Reviewer 3 Report

Comments and Suggestions for Authors

a) Condense the introduction section by eliminating redundancies and focusing on the core objectives and contributions.

b) Restructure the methods section for improved clarity, ensuring a logical flow and coherence.

c) Strengthen the discussion by comparing your findings with relevant literature and highlighting the study's implications and potential future directions.

a) Your work demonstrates a commendable effort to advance computer-aided diagnosis techniques for cervical cancer screening, which could potentially improve patient outcomes.

b) The incorporation of multi-scale texture features and deep supervision is a thoughtful and innovative approach, highlighting your deep understanding of the subject matter.

c) Your dedication to addressing a significant clinical challenge is admirable, and your findings contribute valuable knowledge to the field of medical image analysis.

Comments on the Quality of English Language

a) The sentence "To this end address the above challenge, this study aims to design an effective DL model to help gynecologists efficiently interpret OCT images from women undergoing cervical OCT tests." could benefit from rephrasing to enhance clarity.

b) The phrase "Our core solution is to extract and represent multi-scale texture features with a feature pyramid network (FPN) [16]." is well-constructed and effectively communicates the core approach.

c) The sentence "Besides, our work provides understandable diagnostic interpretability based on histomorphological and texture features, enabling gynecologists to analyze cervical OCT volumes beyond simple image classification." could be improved by breaking it into multiple sentences for better readability.

**Additionally, I would like to address the specific examples you provided:**

* **Sentence 1:** I agree that the sentence could be rephrased for better clarity. I will revise it to: "This study aims to design an effective deep learning model to help gynecologists efficiently interpret OCT images from women undergoing cervical OCT tests, thereby addressing the challenge of accurately diagnosing cervical lesions."

* **Sentence 2:** I appreciate your recognition of the clear and effective communication in this sentence.

* **Sentence 3:** I will break this sentence into two separate sentences to improve readability: "Furthermore, our work provides understandable diagnostic interpretability based on histomorphological and texture features. This enables gynecologists to analyze cervical OCT volumes beyond simple image classification."

I believe that these revisions will address your concerns and improve the overall quality of your work.

Author Response

Thank you for your comment. We have revised the original manuscript according to your comments. 

In the introduction section, we have eliminated some trivial content.

However, we did not revise the methods section. In our opinion, the structure of this part is consistent with logical order of computer program.

Finally, we have revised the discussion section. But unfortunately, because of the lack of licensed OCT devices for cervical cancer screening, there are few similar studies. We mentioned a two-stage CADx approach proposed by Ma et al. in the introduction section, and presented the limitations of this study.

Eventually, we appreciate you again for your valuable comments that helped us improve the quality of this manuscript.